# Integrated Metabolomic and Transcriptomic Analysis Reveals Differential Mechanism of Flavonoid Biosynthesis in Two Cultivars of *Angelica sinensis*

**DOI:** 10.3390/molecules27010306

**Published:** 2022-01-04

**Authors:** Tiantian Zhu, Minghui Zhang, Hongyan Su, Meiling Li, Yuanyuan Wang, Ling Jin, Mengfei Li

**Affiliations:** 1College of Pharmacy, Gansu University of Chinese Medicine, Lanzhou 730101, China; ztt0935@gszy.edu.cn (T.Z.); zmh0819@tom.com (M.Z.); wyy19880821@126.com (Y.W.); 2Northwest Collaborative Innovation Center for Traditional Chinese Medicine, Lanzhou 730000, China; 3State Key Laboratory of Aridland Crop Science, Gansu Agricultural University, Lanzhou 730070, China; Shy922322@163.com (H.S.); mlli1996@163.com (M.L.)

**Keywords:** *Angelica sinensis*, cultivar, flavonoids, anthocyanins, metabolomics, transcriptomics

## Abstract

*Angelica sinensis* is a traditional Chinese medicinal plant that has been primarily used as a blood tonic. It largely relies on its bioactive metabolites, which include ferulic acid, volatile oils, polysaccharides and flavonoids. In order to improve the yield and quality of *A. sinensis*, the two cultivars Mingui 1 (M1), with a purple stem, and Mingui 2 (M2), with a green stem, have been selected in the field. Although a higher root yield and ferulic acid content in M1 than M2 has been observed, the differences of flavonoid biosynthesis and stem-color formation are still limited. In this study, the contents of flavonoids and anthocyanins were determined by spectrophotometer, the differences of flavonoids and transcripts in M1 and M2 were conducted by metabolomic and transcriptomic analysis, and the expression level of candidate genes was validated by qRT-PCR. The results showed that the contents of flavonoids and anthocyanins were 1.5- and 2.6-fold greater in M1 than M2, respectively. A total of 26 differentially accumulated flavonoids (DAFs) with 19 up-regulated (UR) and seven down-regulated (DR) were obtained from the 131 identified flavonoids (e.g., flavonols, flavonoid, isoflavones, and anthocyanins) in M1 vs. M2. A total 2210 differentially expressed genes (DEGs) were obtained from the 34,528 full-length isoforms in M1 vs. M2, and 29 DEGs with 24 UR and 5 DR were identified to be involved in flavonoid biosynthesis, with 25 genes (e.g., *CHS1*, *CHI3*, *F3H*, *DFR*, *ANS*, *CYPs* and *UGTs*) mapped on the flavonoid biosynthetic pathway and four genes (e.g., *RL1*, *RL6*, *MYB90* and *MYB114*) belonging to transcription factors. The differential accumulation level of flavonoids is coherent with the expression level of candidate genes. Finally, the network of DAFs regulated by DEGs was proposed. These findings will provide references for flavonoid production and cultivars selection of *A. sinensis*.

## 1. Introduction

*Angelica sinensis* (Oliv.) Diels (syn. *Angelica polymorpha* Maxim. var. *sinensis* Oliv.) is an Apiaceae (alt. Umbelliferae) perennial rhizomatous species and commonly named as Dang gui, Dong quai and Toki [1]. The species is originally native to China, with a population center in Gansu and widely cultivated at altitudes of 2000 to 3000 m [1,2,3]. The roots were first recorded in the earliest known herbal text “*Shen Nong Ben Cao Jing*”, and have been used as a traditional Chinese medicine for nourishing the blood, regulating female menstrual disorders, relieving pain, and relaxing the bowels, etc., for over 2000 years [4,5]. Recently, the roots have been used as potential treatments of acute ischemic stroke, chronic obstructive pulmonary disease with pulmonary hypertension, as well as for its cardio-cerebrovascular, immunomodulatory and antioxidant effects [5,6,7]. Phytochemical and pharmacological investigations have demonstrated that these therapeutic properties largely rely on bioactive metabolites including ferulic acid, volatile oils, polysaccharides and flavonoids [1,5,8].

In order to improve the yield and quality of *A. sinensis*, strategies undertaken include selecting suitable cultivation areas [3,9], cultivating with standard methods [10,11,12], inhibiting early bolting and flowering [13,14,15], and breeding fine cultivars [16]. Currently, six *A. sinensis* cultivars that are recorded include: Mingui 1 (M1) with a purple stem, Mingui 2 (M2) with a green stem, and Mingui 5, with bigger leaves selected using a systematic breeding method, as well as Mingui 3, Mingui 4 and 6 from the M1 irradiated by heavy ion (55 Mev/u^40^ Ar^17+^) [17]. Among the six cultivars, the M1 dominates in the production with cultivated area at over 41,000 ha (95%). Due to its high yield, the M2 is gradually accepted due to its low rate of early bolting and flowering [16,17,18].

For the difference of bioactive metabolites between M1 and M2, greater ferulic acid content and less ligustilide content has been observed in M1 in comparison with M2 [19,20]. The differences of other bioactive metabolites, including volatile oils, polysaccharides and flavonoids, as well as the stem-color formation between M1 and M2, have not been investigated. In this study, we examined the differences of flavonoids and transcripts based on metabolomic and transcriptomic analysis, and found that the flavonoid and anthocyanin contents were greater in M1 than M2; 26 flavonoids were differentially accumulated; and 29 genes involved in flavonoid biosynthesis were differentially expressed in M1 vs. M2.

## 2. Results

### 2.1. Differenence of Flavonoid and Anthocyanin Contents between the Two Cultivars

As is shown in Figure 1, a significant difference of flavonoid and anthocyanin contents was observed, with a 1.48- and 2.57-fold greater amount in M1 than M2, respectively.

### 2.2. Targeted-Flavonoids Metabolomic Analysis

#### 2.2.1. Identification of Differentially Accumulated Flavonoids (DAFs) in M1 vs. M2

A total of 131 flavonoids that were identified by LC-ESI-MS/MS analysis include flavonols (51), flavonoid (36), flavanols (8), chalcones (7), dihydroflavone (7), dihydroflavonol (7), isoflavones (6), anthocyanins (5), and flavonoid carbonoside (4) (Figure 2). Among them, 26 (19 UR and 7 DR) flavonoids that were differentially accumulated in M1 vs. M2 based on the principal component analysis (PCA) and orthogonal projection to latent structures-discriminant analysis (OPLS-DA) (Appendix A). The specific flavonoids and their differential accumulation levels in M1 vs. M2 are shown in Table 1 and Appendix A.

#### 2.2.2. Pathway Enrichment of DAFs

Among the 26 DAFs, seven metabolites were enriched in five pathways, including anthocyanin biosynthesis (cyanidin-3-*O*-glucoside and cyanidin-3-*O*-sambubioside, ko00942), flavone and flavonol biosynthesis (quercetin-3-*O*-rhamnoside, quercetin-3-*O*-glucoside and quercetin-3-*O*-sambubioside, ko00944), metabolic pathways (quercetin-3-*O*-glucoside, ko01100), flavonoid biosynthesis (naringenin-7-*O*-glucoside and isosalipurposide, ko00941), and biosynthesis of secondary metabolites (quercetin-3-*O*-glucoside, ko01110) based on the Kyoto Encyclopedia of Genes and Genomes (KEGG) databases analysis (Figure 3).

### 2.3. Isoforms Analysis

A total of 702,133 high-fidelity reads were extracted after 38 full passes of raw reads (Appendix A), 45,026 polished high-quality isoforms were obtained using a Quiver calculation (Appendix A), and 34,528 full-length isoforms were generated after the full-length non-chimeric (FLNC) reads clustered and integrated (Appendix A).

The 34,528 isoforms were annotated against the KEGG (33,241), KOG (22,601), Nr (33,947) and SwissProt (29,150) databases (Figure 4A), and the top 10 species distribution includes *Daucus carota*, *Actinidia chinensis*, *Prunus dulcis*, *Camellia sinensis*, *Carica papaya*, *Angelica sinensis*, *Petroselinum crispum*, *Vitis vinifera*, *Brassica napus*, and *Artemisia annua* (Figure 4B).

### 2.4. Transcriptomic Analysis between M1 and M2

#### 2.4.1. Global Gene Analysis

To reveal molecular mechanisms responsible for the difference of flavonoid accumulation and the stem-color formation, comparison of gene transcription between M1 and M2 was performed. After data filtering, 38.65 and 38.73 million high-quality reads were collected, and 27.92 and 28.36 multiple mapped reads were obtained from the M1 and M2, respectively. Meanwhile, the exon rate reached 100% (Table 2).

#### 2.4.2. Identification of Differentially Expressed Genes (DEGs)

A total of 2210 DEGs were observed from the 34,528 full-length isoforms, with 1110 UR and 1100 DR in M1 vs. M2 (Figure 5A), based on the Reads Per kb per Million (RPKM) value (Appendix A), PCA (Appendix A) and Pearson correlation analysis (Appendix A). The cluster heat map of the 2,210 DEGs was shown in Figure 5B.

#### 2.4.3. Functional Annotation and Enrichment of DEGs

The function of the 2210 DEGs was annotated against the Gene Ontology (GO) and KO databases. For the GO database, 48 terms were classified into biological process (22), cellular component (16), and molecular function (10) (Appendix A). For the KO database, 1784 DEGs were enriched 103 pathways, with top 10 pathways including: oxidative phosphorylation; metabolic pathways; linoleic acid metabolism; ABC transporters; alpha-Linolenic acid metabolism; nitrogen metabolism; phenylpropanoid biosynthesis; TCA cycle; cutin, suberine and wax biosynthesis; and pyruvate metabolism (Figure 6).

### 2.5. DEGs Involved in Flavonoids Biosynthesis

Twenty-nine DEGs (24 UR and 5 DR) were identified to be involved in flavonoid biosynthesis. Twenty-five genes were mapped on flavonoid biosynthetic pathway with a 1.04 to 8.63-fold UR for the 20 genes *CHS1*, *CHI3*, *F3H*, *DFR*, *ANS*, *CGT*, *GT6*, *UGT85A8*, *F3GT1*, *P5MaT*, *CYP71A1*, *CYP71A9, CYP71D313*, *CYP71B26*, *CYP71B36*, *CYP72A219*, *CYP736A12*, *CYP76AD1*, *CYP76A2* and *CYP77A3*, and a −1.05 to −1.52-fold DR for the 5 genes *UGT73C6*, *3MaT*, *CYP71B34*, *CYP71B35* and *CYP81Q32* (Table 3). Four genes belonged to transcription factors (TFs) with a 3.76-, 1.15-, 1.19- and 2.40-fold UR for *RL1*, *RL6*, *MYB90* and *MYB114*, respectively (Table 4).

### 2.6. Network of DAFs Regulated by DEGs

The 26 DAFs and 25 DEGs (exclude 4 TFs) were connected based on the flavonoid biosynthetic pathway analyzed by KO enrichment and biological function of proteins on the SwissProt database, and the proposed biosynthetic pathway is shown in Figure 7. Flavonoids are synthesized via the phenylpropanoid pathway. Briefly, the upstream metabolite 4-coumaroyl-CoA is formed from phenylalanine by the catalyzation of PAL, C4H and 4CL. The 4-coumaroyl-CoA is converted into two metabolites, isoliquiritigenin and naringenin chalcone, by the catalyzation of CHS, then respectively transformed to liquiritigenin and naringenin by the catalyzation of CHI. In the sub-pathway of isoflavonoid biosynthesis, 13 cytochrome P450 monooxygenases (CYPs) are involved, and butin-7-*O*-glucoside (21) is produced by the catalyzation of F3′H. In the sub-pathway of anthocyanin biosynthesis, 10 genes (*F3H*, *DFR*, *ANS*, *CGT*, *GT6*, *UGT85A8*, *UGT73C6*, *F3GT1*, *3MaT* and *P5MaT*) and four anthocyanins are involved, and pelargonidin-3-*O*-glucoside-5-*O*-arabinoside (23), cyanidin-3-*O*-glucoside (24), cyanidin-3-*O*-sambubioside (25) and peonidin-3-*O*-sambubioside (26) are formed. Under the catalyzation of FLS and F3′H, the metabolites kaempferol and quercetin are produced, then 2 kaempferol-derivatives (18 and 19) and 14 quercetin-derivatives (1 to 14) are formed. In addition, the chrysoeriol-5-*O*-glucoside (15), naringenin-7-*O*-glucoside (16), hesperetin-5-*O*-glucoside (17) and isosalipurposide (20) are also mapped in the phenylpropanoid pathway.

### 2.7. qRT-PCR Validation of Candidate Genes Involved in Flavonoid Biosynthesis

As shown in Figure 7, 25 DEGs were mapped in the pathway of flavonoid biosynthesis, with 20 UR and 5 DR in M1 vs. M2 (Table 3). Among them, 22 genes (20 UR and 2 DR) were selected to qRT-PCR validation, and their relative expression levels (RELs) were consistent with RPKM values (Table 3, Figure 8).

Specifically, five genes directly participating in upstream flavonoid biosynthesis included *CHS1*, *CHI3*, *F3H*, *DFR* and *ANS*, their RELs exhibited a 31.70-, 1.21-, 10.63-, 14.24- and 26.00-fold UR, respectively, in M1 vs. M2 (Figure 8A).

Thirteen CYP genes participate in isoflavonoid biosynthesis with 10 UR genes including *CYP71A1*, *CYP71A9*, *CYP71D313*, *CYP71B26*, *CYP71B36*, *CYP72A219*, *CYP736A12*, *CYP76AD1*, *CYP76A2* and *CYP77A3*, as well as 3 DR genes including *CYP71B34*, *CYP71B35* and *CYP81Q32* (Figure 7), The RELs of the 10 UR genes exhibited a 4.09-, 3.58-, 8.47-, 5.45-, 8.16-, 6.86-, 5.61-, 14.47-, 3.67- and 3.81-fold, respectively, in M1 vs. M2 (Figure 8B).

Seven genes directly participating in anthocyanin biosynthesis included *CGT*, *GT6*, *UGT85A8*, *UGT73C6*, *F3GT1*, *3MaT* and *P5MaT*. The RELs of the five genes *CGT*, *GT6*, *UGT85A8*, *F3GT1* and *P5MaT* exhibited a 6.83-, 8.58-, 3.84-, 28.86- and 5.33-fold UR, while the two genes *UGT73C6* and *3MaT* exhibited a 0.54- and 0.68-fold DR, respectively, in M1 vs. M2 (Figure 8C).

Four TFs participating in regulating anthocyanin biosynthesis included *RL1*, *RL6*, *MYB90* and *MYB114*, their RELs exhibited a 35.92-, 4.54-, 7.90-, and 3.75-fold UR, respectively, in M1 vs. M2 (Figure 8D).

## 3. Discussion

Accumulation of secondary metabolites is not only influenced by environmental factors (e.g., temperature, light, the supply of water and minerals) but also genotypes (e.g., variety, strain and cultivar) [21,22,23]. Previous studies have demonstrated that there is a significant difference of secondary metabolites among the three *Angelica* species: *A. sinensis* (Oliver) Diels, *A. dahurica* (Fisch. ex Hoffn) Benth, et Hook. F, and *A. pubescens* Maxim [5]. A higher root yield of M1 than M2 was observed [16,17,18]. In this study, the flavonoid and anthocyanin contents were greater in M1 than M2, 26 DAFs (19 UR and 7 DR) and 29 DEGs (24 UR and 5 DR) involved in flavonoid biosynthesis were observed in M1 vs. M2.

Flavonoids are widely distributed secondary metabolites with different metabolic functions in plants, such as providing colors attractive to plant pollinators, promoting physiological survival, and protecting plants from fungal pathogens and UV-B radiation; meanwhile, flavonoids possess antifungal, antioxidant and anticancer activities [24,25]. In this study, a 1.48 increase of flavonoid content was observed in M1 compared to M2 (Figure 1), suggesting that the adaptation ability of M1 to environmental conditions is stronger than that of M2, which is consistent with previous investigations that the yield and tolerance of M1 is greater and stronger than that of M2 in the field [16,17,18]. Additionally, a 2.57-fold greater anthocyanin content was observed in M1 compared to M2 (Figure 1), which can describe the difference of stem-color formation for M1 with purple stem and M2 with green stem. Several studies have reported that the contents of flavonoids and anthocyanins play a positive role in pigmentation [26,27].

Currently, more than 6000 different flavonoids have been identified from plants, and they can be classified into six major subgroups, including chalcones, flavones, flavonols, flavandiols, anthocyanins, and proanthocyanidins or condensed tannins [28]. In this study, 131 flavonoids were identified from M1 and M2 by LC-ESI-MS/MS analysis including flavonols (51), flavonoid (36), flavanols (8), chalcones (7), dihydroflavone (7), dihydroflavonol (7), isoflavones (6), anthocyanins (5), and flavonoid carbonoside (4); and 26 of them were differentially accumulated in M1 vs. M2 (Figure 2 and Table 1). The 26 DAFs were enriched in five pathways and mapped on the phenylpropanoid pathway (Figure 3, Figure 7).

Extensive experiments have demonstrated that the expression of genes encoding enzymes and TFs is responsible for the formation of flavonoid structures and their subsequent modification reactions [29]. In this study, 29 genes participating in flavonoid biosynthesis were screened from the 2210 DEGs in M1 vs. M2; the specific role of the 29 genes has been linked with the 26 DAFs and mapped on the phenylpropanoid pathway (Figure 5 and Figure 7; Table 3 and Table 5).

Flavonoids are synthesized via the phenylpropanoid pathway with transforming phenylalanine into 4-coumaroyl-CoA, and then entering the sub-pathways of flavonoid biosynthesis under the coordinated regulation of key genes (Figure 7). In this study, five genes that directly participate in upstream flavonoid biosynthesis include *CHS1*, *CHI3*, *F3H*, *DFR* and *ANS*, which is responsible for sequentially converting 4-coumaroyl-CoA to naringenin chalcone, naringenin, dihydrokaempferol, leucoanthocyanidin and pelargonidin [30,31,32].

CYPs play diverse roles in metabolism including the synthesis of secondary metabolites (e.g., flavonoids, alkaloids and lignan) [33,34]. Previous studies have found that the overexpression of *CYPs* genes promotes flavonoid and pigment biosynthesis [35,36]. In this study, 13 *CYPs* genes directly participate in isoflavonoid biosynthesis with 10 UR (Figure 7; Table 3), which will enhance the flavonoid biosynthesis and greater accumulation in M1 compared to M2 (Figure 1).

UDP-glycosyltransferases (UGTs) is one of the glycosyltransferases that comprise a highly divergent and polyphyletic multigene family involved in widespread glycosylation of plant secondary metabolites (e.g., anthocyanins) [37]. In this study, five *UGTs* genes were observed to participate in anthocyanin biosynthesis. Previous studies have found that *CGT* is involved in the biosynthesis of mangiferin [38], *GT6* is involved in the formation of flavonol 3-*O*-glucosides [39], *UGT85A8* is involved in glycosylate diterpenes or flavonols, *UGT73C6* is involved in flavonol biosynthetic process while possessing low quercetin 3-*O*-glucosyltransferase, 7-*O*-glucosyltransferase and 4′-*O*-glucosyltransferase activities [40,41], and *F3GT1* is involved in anthocyanin biosynthesis by catalyzing the galactosylation of cyanidin [42]. Meanwhile, two genes encoding malonyltransferase (MaT) that is also involved in anthocyanin biosynthesis include: *3MaT* involved in the transfer of the malonyl group from malonyl-CoA to pelargonidin 3-*O*-glucoside to produce pelargonidin 3-*O*-6″-*O*-malonylglucoside [43], and *P5MaT* involved in the transfer of the malonyl group from malonyl-CoA to the 4‴-hydroxyl group of the 5-glucosyl moiety of anthocyanins [44]. 

TFs play a great role in controlling cellular processes and MYB TF family is involved in controlling various processes such as responses to biotic and abiotic stresses, development, and metabolism, etc [45]. Several investigations have found that the overexpression of MYB TFs promote flavonoid and anthocyanin biosynthesis [46,47]. In this study, 4 MYB TFs were observed to be in involved in anthocyanin biosynthesis. The two TFs *RL1* and *RL6* as a member of the MYB-related gene family may regulate the anthocyanin biosynthesis [48]. The two TFs MYB90 and *MYB114* are transcription activators, when associated with BHLHs/MYCs, EGL3, or GL3, they promote the synthesis of phenylpropanoid-derived compounds such as anthocyanins [49,50].

## 4. Materials and Methods

### 4.1. Plant Material

Functional leaves and petioles of two-year-old *Angelica sinensis* [two cultivars: Mingui 1 (M1) with purple stem and Mingui 2 (M2) with green stem, Appendix A] were collected from the city-owned breeding garden located in Shangconggou village, Huichuan Town, Weiyuan County, Dingxi City (2507 m a.s.l.; 35°2′39″ N, 104°1′55″ E) of Gansu province, China in July 2020. The two cultivars were identified by Professor Ling Jin (Gansu University of Chinese Medicine, Lanzhou, China). Voucher specimens (M1: 20190725GSWYMG1, M2: 20190725GSWYMG2) were deposited in the herbarium of College of Pharmacy, Gansu University of Chinese Medicine, Lanzhou, China. During the growth stages, the two cultivars were maintained with the same field management conditions. The collected samples (leaves and petioles = 1:1, g/g fresh weight; n = 20 plants) were immediately frozen in liquid nitrogen for total flavonoid and anthocyanin measurement, metabolomic and transcriptomic analysis.

### 4.2. Chemicals

Standards of metabolites used for UPLC analysis were purchased from BioBioPha (Kunming, Yunnan, China) and Sigma-Aldrich (St Louis, MO, CA, USA). All chemicals and reagents (e.g., AlCl_3_, catechin, ethanol, HCL, methanol, NaNO_2_ and NaOH) were of analytical grade and purchased from Merck, Germany. Trizol reagent, RT Kit and SuperReal PreMix were purchased from Tiangen, China.

### 4.3. Measurement of Total Flavonoid and Anthocyanin Contents

#### 4.3.1. Measurement of Total Flavonoid Content

Fresh samples (0.5 g) were placed in ethanol (5 mL, 95% *v*/*v*) and ground, the homogenate was centrifuged at 5000 r/min for 10 min at 4 °C and re-extracted twice more. The extracts were increased to 20 mL with ethanol (95% *v*/*v*). Total flavonoids content was measured according to a NaNO_2_-AlCl_3_-NaOH method [51,52]. Briefly, extracts (150 μL) were added in ddH_2_O (2 mL) and NaNO_2_ (5% *w*/*v*, 0.3 mL). After the mixture agitating for 5 min, AlCl_3_ (10% *w*/*v*, 0.3 mL) was added and reacted for 1 min, then NaOH (1.0 mol/L, 2 mL) was added to stop the reaction. Absorbance readings were taken at 510 nm using a spectrometer. Total flavonoid content was calculated based on a standard curve and expressed as mg of catechin.

#### 4.3.2. Measurement of Anthocyanin Content

Anthocyanins content was measured according to a previous protocol [53]. Fresh samples (0.5 g) were placed in methanol (5 mL, 0.1% HCL *v*/*v*) and ground, the homogenate was centrifuged at 5000 r/min for 30 s at 4 °C and re-extracted twice more. The extracts were increased to 20 mL with methanol (0.1% HCL *v*/*v*). Absorbance readings were taken at 530 nm using a spectrometer. Anthocyanins content was evaluated based on a relative expression level compared to the blank control.

### 4.4. Metabolomic Analysis

#### 4.4.1. Sample Preparation and Extraction

The freeze-dried samples were ground in a mixer mill with zirconia beads for 1.5 min at 30 Hz. The powder (0.1 g) was added into methanol (70% *v*/*v*, 1 mL) and extracted for 12 h at 4 °C, and then the homogenate was centrifuged at 10,000 r/min for 10 min at 4 °C. The supernatant was filtrated with a 0.22 μm durapore membrane for LC–MS/MS analysis.

#### 4.4.2. UPLC Analysis

The metabolites were firstly analyzed using a LC-ESI-MS/MS system (UPLC, Shim-pack UFLC CBM-20A, Shimadzu, Japan). Extracts (5 μL) were analyzed using a Waters ACQUITY UPLC HSS T3 C18 column (100 mm × 2.1 mm, 1.8 μm; m; column temperature 40 °C). Acetic acid (0.04% *v*/*v*, A)—acetonitrile (B) made up the mobile phase with gradient elution: 5% B (0–11 min), 95% B (11–12 min) and 5% B (12.1–15 min) at a flow rate of 0.4 mL/min. Quality control samples were mixed by all the samples to detect reproducibility of the whole experiments (Appendix A).

#### 4.4.3. MS/MS Analysis

The effluent from UPLC was analyzed using an AB SCIEX QTRAP 4500 and Triple Quad 4500 Systems (AB SCIEX, Boston, MA, USA) equipped with an ESI-Turbo Ion-Spray interface and operated in a positive ion mode. The operation parameters were as follows: ESI source temperature 550 °C, ion spray voltage 5500 V, curtain gas 25 psi, collision-activated dissociation set 5 pis. Triple quadrupole scans were acquired as MRM experiments with optimized de-clustering potential and collision energy CE for each individual multiple reaction monitoring (MRM) transitions. The *m*/*z* range was set between 50 and 1000.

#### 4.4.4. Metabolites Identification

Metabolites were identified using internal and public databases (MassBank, KNApSAcK, HMDB, MoTo DB and METLIN) and comparing *m*/*z* values, retention time, and the fragmentation patterns with the standards. 

#### 4.4.5. Differential Metabolites Analysis

The accumulation level of metabolites was ranked using a variable importance in projection (VIP) scores of orthogonal projection to latent structures-discriminant analysis (OPLS-DA). The level of differential accumulation between M1 and M2 was determined with a criterion of VIP ≥ 1 and *t*-test *p* ≤ 0.05.

### 4.5. Isoform Sequencing and Transcriptomic Analysis

#### 4.5.1. cDNA Library Construction and Single Molecular Real-Time (SMRT) Sequencing

Total RNA was extracted using a Trizol reagent according to the manufacturer’s protocol. The quality of extracted RNA was determined using a Agilent 2100 Bioanalyzer and agarose gel electrophoresis. mRNA was enriched by Oligo (dT) magnetic beads and transcribed into cDNA using a Clontech SMARTer cDNA Synthesis Kit. Then the cDNA was amplified by PCR for 13 cycles to prepare for the next SMRTbell library construction. The > 5 kb size sequence was ligated to the sequencing adapters. The SMRTbell template was applied and sequenced on a PacBio SequelII platform (Gene Denovo Biotechnology Co., Ltd., Guangzhou, China).

#### 4.5.2. Isoform Data Processing

The raw sequencing reads of cDNA libraries were analyzed using a isoform sequencing (Iso-Seq) system [54]. Briefly, high quality CCS were extracted and then the FLNC reads were obtained after removing the primers, barcodes, poly (A) tail trimmings and concatemers. The FLNC reads were clustered to generate the entire isoform, which was used for sequences correction. Finally, isoforms were BLAST analyzed against the Nr database, isoforms were annotated against the databases including: KEGG, KOG and Swiss-Prot.

#### 4.5.3. Transcriptomic Analysis and DEGs Identification

Total RNA was extracted using a Trizol reagent according to the manufacturer’s protocol. The processes of enrichment by Oligo (dT) magnetic beads, fragmentation by ultrasonic, reverse transcription by a cDNA Synthesis Kit, synthesis of the second-strand cDNA by PCR amplification as well as purification of cDNA fragments by end-repairing and adapter-connecting were conducted according to previous protocols [55]. RNA-seq was performed by an Illumina HiSeqTM 4000 platform (Gene Denovo Biotechnology Co., Ltd., Guangzhou, China). Raw reads were filtered according to previous protocols (Li et al., 2008). Clean reads was assembled using Trinity [56]. The expression level of each transcript was normalized to RPKM values [57], and the differential expression level between M1 and M2 was determined with a criteria of |log_2_ (fold-change)| ≥ 1 and *p* ≤ 0.05 by DESeq2 software and the edgeR package [58,59].

### 4.6. qRT-PCR Validation of Genes Involved in Flavonoid Biosynthesis

The primer sequence (Table 5) was designed via a primer-blast in NCBI and synthesized by reverse transcription (Sangon Biotech Co., Ltd., Shanghai, China). First cDNA was synthesized using a RT Kit. PCR amplification was performed using a SuperReal PreMix. Melting curve was analyzed at 72 °C for 34 s. *Actin* gene was used as a reference control. The RELs of gene were calculated using a 2^−ΔΔCt^ method [60].

### 4.7. Statistical Analysis

All experiments were performed in three biological replicates in this study. A *t*-test in SPSS 22.0 was performed for independent treatments with *p* < 0.05 as the basis for statistical differences.

## 5. Conclusions

From the above observations, the flavonoid and anthocyanin contents in the cultivar M1 of *A. sinensis* were greater than in M2, which rely on the up-regulation of genes involved in flavonoid and anthocyanin biosynthesis. The difference of stem color formation between M1 and M2 results from the anthocyanin differential accumulation as well as the genes’ differential expression.

## Figures and Tables

**Figure 1 molecules-27-00306-f001:**
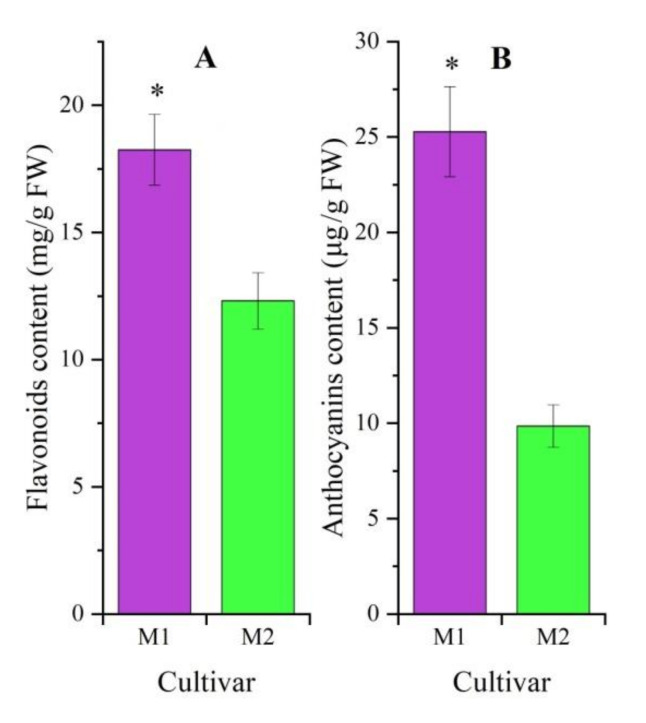
Contents of flavonoids (**A**) and anthocyanins (**B**) in Mingui 1 (M1) and Mingui 2 (M2) (mean ± SD, *n* = 3). A *t*-test was performed for independent treatments, and the “*” is considered significant at *p* < 0.05 between M1 and M2.

**Figure 2 molecules-27-00306-f002:**
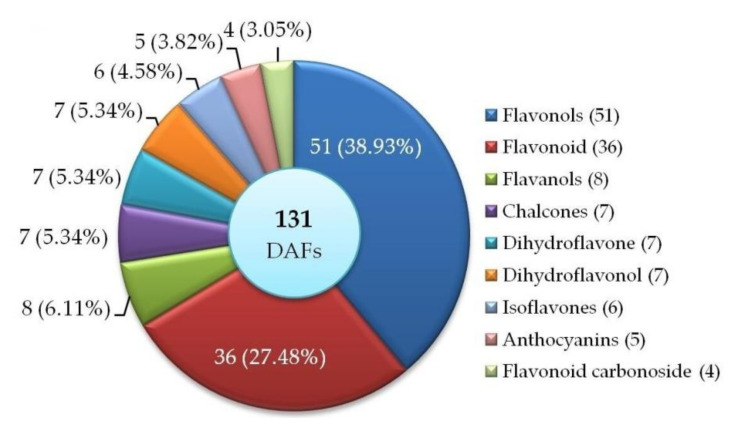
Distribution and classification of DAFs in M1 vs. M2.

**Figure 3 molecules-27-00306-f003:**
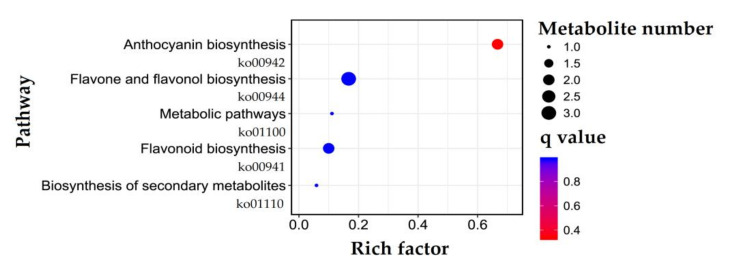
KEGG Orthology (KO) enrichment of the DAFs.

**Figure 4 molecules-27-00306-f004:**
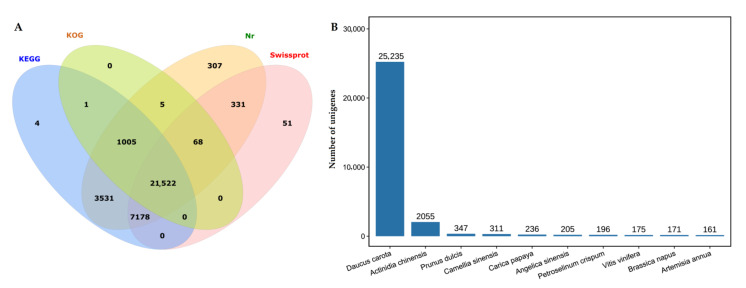
Basic annotation of the isoforms based on KEGG, KOG, Nr and SwissProt databases (**A**) and the top 10 species distribution against Nr (**B**).

**Figure 5 molecules-27-00306-f005:**
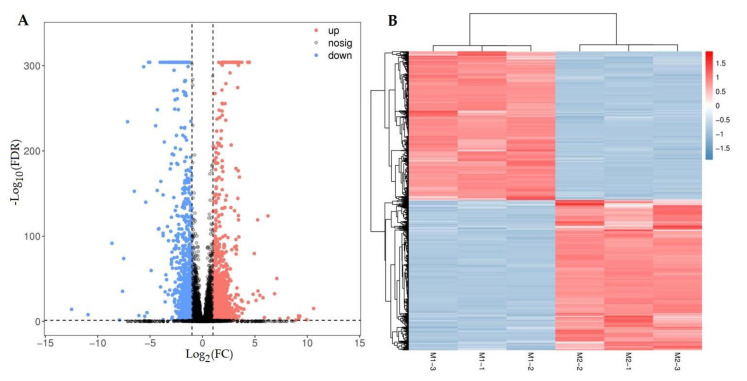
Volcano plot of differential comparison (**A**) and cluster heat map of the DEGs (**B**) in M1 vs. M2.

**Figure 6 molecules-27-00306-f006:**
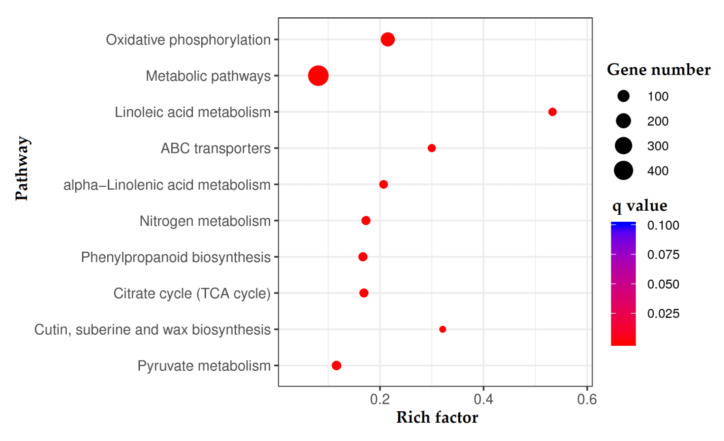
Top 10 pathways of KO enrichment of the DEGs.

**Figure 7 molecules-27-00306-f007:**
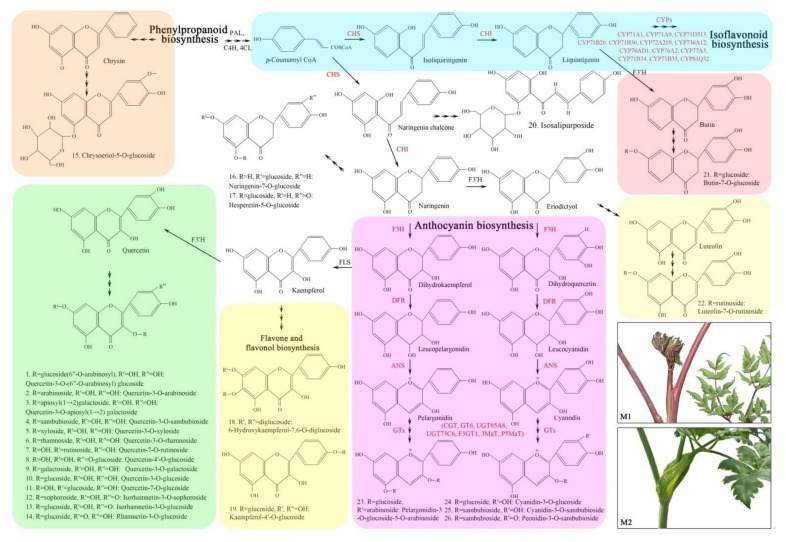
Proposed network of the DAFs regulated by the DEGs in M1 and M2. The 26 DAFs are listed from No.1 to No.26 (Table 1), and the enzymes encoded by the 25 DEGs are colored in red.

**Figure 8 molecules-27-00306-f008:**
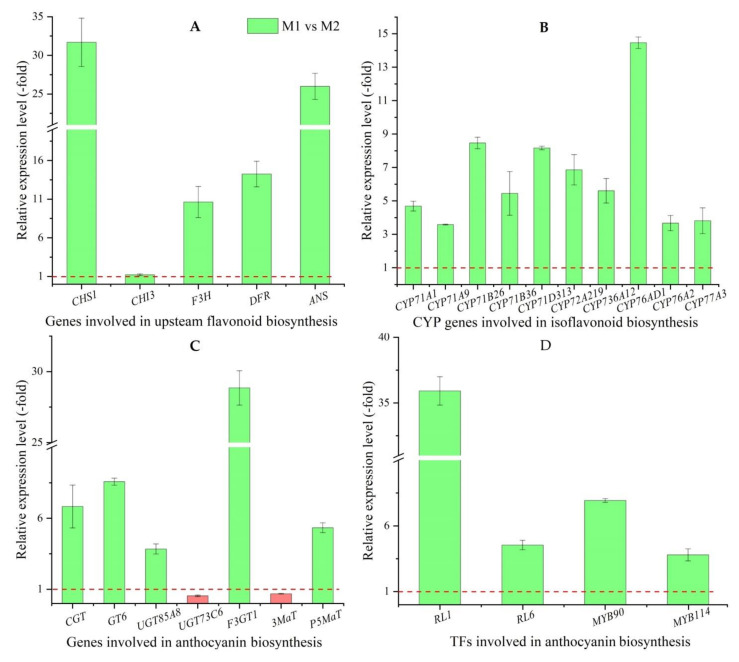
The RELs of genes involved in flavonoid biosynthesis (**A**), isoflavonoid biosynthesis (**B**), anthocyanin biosynthesis (**C**), and TFs (**D**) in M1 vs. M2, as determined by qRT-PCR (mean ± SD, *n* = 3). The column highlighted in green represents genes favoring flavonoid biosynthesis and red represents genes disfavoring flow.

**Table 1 molecules-27-00306-t001:** Classification of DAFs and their differential accumulation levels in M1 vs. M2 (mean ± SD, *n* = 3).

No.	Compounds Name	Formula	log_2_FC(M1 vs. M2)
1	Quercetin-3-*O*-(6″-*O*-arabinosyl) glucoside	C_26_H_28_O_16_	0.90 ± 0.12
2	Quercetin-3-*O*-arabinoside (Guaijaverin)	C_20_H_18_O_11_	0.65 ± 0.07
3	Quercetin-3-*O*-apiosyl(1→2) galactoside	C_26_H_28_O_16_	0.65 ± 0.06
4	Quercetin-3-*O*-sambubioside	C_26_H_28_O_16_	0.55 ± 0.01
5	Quercetin-3-*O*-xyloside (Reynoutrin)	C_20_H_18_O_11_	0.35 ± 0.07
6	Quercetin-3-*O*-rhamnoside (Quercitrin)	C_21_H_20_O_11_	0.34 ± 0.06
7	Quercetin-7-*O*-rutinoside	C_27_H_30_O_16_	0.21 ± 0.02
8	Quercetin-4′-*O*-glucoside (Spiraeoside)	C_21_H_20_O_12_	−1.02 ± 0.11
9	Quercetin-3-*O*-galactoside (Hyperin)	C_21_H_20_O_12_	−0.93 ± 0.06
10	Quercetin-3-*O*-glucoside (Isoquercitrin)	C_21_H_20_O_12_	−0.78 ± 0.06
11	Quercetin-7-*O*-glucoside	C_21_H_20_O_12_	−0.66 ± 0.14
12	Isorhamnetin-3-*O*-sophoroside	C_28_H_32_O_17_	0.63 ± 0.07
13	Isorhamnetin-3-*O*-Glucoside	C_22_H_22_O_12_	0.38 ± 0.01
14	Rhamnetin-3-*O*-Glucoside	C_22_H_22_O_12_	0.38 ± 0.01
15	Chrysoeriol-5-*O*-glucoside	C_22_H_22_O_11_	−2.17 ± 0.21
16	Naringenin-7-*O*-glucoside (Prunin)	C_21_H_22_O_10_	1.15 ± 0.16
17	Hesperetin-5-*O*-glucoside	C_22_H_24_O_11_	−0.86 ± 0.11
18	6-Hydroxykaempferol-7,6-*O*-Diglucoside	C_27_H_30_O_17_	0.47 ± 0.06
19	Kaempferol-4′-*O*-glucoside	C_21_H_20_O_11_	−0.39 ± 0.05
20	Isosalipurposide (Phlorizin Chalcone)	C_21_H_22_O_10_	1.56 ± 0.24
21	Butin-7-*O*-glucoside	C_21_H_22_O_10_	0.95 ± 0.21
22	Luteolin-7-*O*-rutinoside	C_27_H_30_O_15_	0.37 ± 0.06
23	Pelargonidin-3-*O*-glucoside-5-*O*-arabinoside	C_26_H_29_O_14_+	18.66 ± 1.22
24	Cyanidin-3-*O*-glucoside (Kuromanin)	C_21_H_21_O_11_+	19.15 ± 1.73
25	Cyanidin-3-*O*-sambubioside	C_26_H_29_O_15_+	8.85 ± 1.09
26	Peonidin-3-*O*-sambubioside	C_27_H_31_O_15_+	6.26 ± 0.85

Note: The level of differential accumulation between M1 and M2 was determined with a criterion of variable importance in projection (VIP) ≥ 1 and *t*-test *p* ≤ 0.05.

**Table 2 molecules-27-00306-t002:** Summary of sequencing data of M1 and M2 (mean ± SD, *n* = 3).

	M1	M2
Filtered data		
Data of reads number (million)	38.65 ± 1.34	38.73 ± 1.90
Data of reads number × read length (million)	5773.93 ± 2.00	5784.44 ± 2.83
Q20(%)	97.82 ± 0.03	98.09 ± 0.21
Q30(%)	93.39 ± 0.08	94.05 ± 0.54
Mapped data against full-length isoforms		
Data of unique mapped reads (million)	6.21 ± 0.19	6.25 ± 0.29
Data of multiple mapped reads (million)	27.92 ± 0.87	28.36 ± 1.35
Mapping ratio (%)	88.32 ± 0.31	89.37 ± 0.17
Exon rate (%)	100	100

**Table 3 molecules-27-00306-t003:** DEGs involved in flavonoid biosynthesis and their RPKM values in M1 vs. M2.

Gene Name	Protein Name	SwissProt ID	log_2_FC(M1 vs. M2)
*CHS1*	Chalcone synthase 1	Q9ZS41	8.63
*CHI3*	Probable chalcone--flavonone isomerase 3	Q8VZW3	1.06
*F3H*	Flavanone 3-dioxygenase	Q7XZQ7	1.97
*DFR*	Dihydroflavonol 4-reductase	P51105	6.50
*ANS*	Leucoanthocyanidin dioxygenase	P51091	7.51
*CGT*	UDP-glycosyltransferase 13	A0A0M4KE44	1.06
*GT6*	UDP-glucose flavonoid 3-*O*-glucosyltransferase 6	Q2V6K0	2.25
*UGT85A8*	UDP-glycosyltransferase 85A8	Q6VAB3	1.31
*UGT73C6*	UDP-glycosyltransferase 73C6	Q9ZQ95	−1.52
*F3GT1*	Anthocyanidin 3-*O*-galactosyltransferase F3GT1	A0A2R6Q8R5	1.17
*3MaT*	Malonyl-coenzyme A:anthocyanin 3-*O*-glucoside-6″-*O*-malonyltransferase	Q8GSN8	−1.28
*P5MaT*	Pelargonidin 3-*O*-(6-caffeoylglucoside) 5-*O*-(6-*O*-malonylglucoside) 4‴-malonyltransferase	Q6TXD2	1.21
*CYP71A1*	Cytochrome P450 71A1	P24465	1.19
*CYP71A9*	Cytochrome P450 71A9	O81970	1.04
*CYP71D313*	Cytochrome P450 CYP71D313	H2DH20	2.21
*CYP71B26*	Cytochrome P450 71B26	Q9LTL0	1.77
*CYP71B36*	Cytochrome P450 71B36	Q9LIP4	1.33
*CYP72A219*	Cytochrome P450 CYP72A219	H2DH21	1.21
*CYP736A12*	Cytochrome P450 CYP736A12	H2DH18	1.37
*CYP76AD1*	Cytochrome P450 76AD1	I3PFJ5	2.59
*CYP76A2*	Cytochrome P450 76A2	P37122	1.15
*CYP77A3*	Cytochrome P450 77A3	O48928	1.79
*CYP71B34*	Cytochrome P450 71B34	Q9LIP6	−1.05
*CYP71B35*	Cytochrome P450 71B35	Q9LIP5	−1.35
*CYP81Q32*	Cytochrome P450 81Q32	W8JMU7	−1.12

**Table 4 molecules-27-00306-t004:** Differentially expressed transcription factor (TF) involved in flavonoid biosynthesis and their RPKM values in M1 vs. M2.

Gene Name	Protein Name	SwissProt ID	log_2_ FC(M1 vs. M2)
*RL1*	Protein RADIALIS-like 1	F4JVB8	3.76
*RL6*	Protein RADIALIS-like 6	Q1A173	1.15
*MYB90*	Transcription factor MYB90	Q9ZTC3	1.19
*MYB114*	Transcription factor MYB114	Q9FNV8	2.40

**Table 5 molecules-27-00306-t005:** Primer sequence of candidate genes used for qRT-PCR validation.

Gene Name	Primer Sequences (5′ to 3′)	Amplicon Size (bp)
*ACT*	Forward: TGGTATTGTGCTGGATTCTGGT	109
	Reverse: TGAGATCACCACCAGCAAGG	
**Flavonoid biosynthesis (22)**		
*CHS1*	Forward: CATTTCGGGGGCCTAACGAT	197
	Reverse: CCCAACCTCCCGAAGATGAC	
*CHI3*	Forward: CACGGACATTGAGATACACTTCC	111
	Reverse: TCTCCAGTTTTTCCCTTCCAGT	
*F3H*	Forward: AGTGAGAAGTTGATGGCGCT	160
	Reverse: GTCCCAGTGTCAAGTCAGGT	
*DFR*	Forward: ACAGCACTATCACCGCTCAC	134
	Reverse: ATGTATCTTCCCTGCGCTGT	
*ANS*	Forward: GGCCTCAAGTGCCTACAGTT	169
	Reverse: TGTCCAGCCACTCTAACACG	
*CGT*	Forward: GCAGCCCGCAAAATCTGTAG	163
	Reverse: ACGCAACCCTTCCTTGTCTT	
*GT6*	Forward: GTGCCACAGGTGACGATTCT	173
	Reverse: ACTCCCAGTCCCAACTCCTT	
*UGT85A8*	Forward: ATGCAGTATCGCCAACTCGT	111
	Reverse: GTCTTTCATTCCAGGAGCCCA	
*UGT73C6*	Forward: GTATGGGCAGTAAGGGCTGG	110
	Reverse: GCCCAACCACGGATCAAAAG	
*F3GT1*	Forward: GCTTTGGAACTGTGGCGATG	165
	Reverse: AGGCCACGATTTTTCCGGTT	
*3MaT*	Forward: CTCCGTGACATCTCTGCCTC	175
	Reverse: AGCCAACGGAGTGAAGTGTT	
*P5MaT*	Forward: AGGCGAAAAAGGGGTGGAAT	193
	Reverse: GCACCAGTCGGTAAACAAGC	
*CYP71A1*	Forward: GTTTACGTGAGTGCATGGGC	138
	Reverse: TGCCCCAAAAGGAACCAACT	
*CYP71A9*	Forward: CAATGCTTGGGCAACAAACG	153
	Reverse: TTTCTGCTTCTCGGATAGGGC	
*CYP71D313*	Forward: GCTTGGTGAGATCCCTCTGG	108
	Reverse: TCACCAAGTACAAGTCCTGGC	
*CYP71B26*	Forward: TGTTGTGTGGGCCATGACTT	157
	Reverse: TCTCATTGCCTCCTTCACCAC	
*CYP71B36*	Forward: GGGCTGAGAACAGGTCAAGT	199
	Reverse: CTTGTATCGGCTCCTGCAAC	
*CYP72A219*	Forward: TTGCTCGTGTGGACTGTTGT	186
	Reverse: TCGTAGAAGCATACCTGCCG	
*CYP736A12*	Forward: GGAAACCTCCCTCATCGCTC	167
	Reverse: GCCTCAAATTCTGGACGGCT	
*CYP76AD1*	Forward: AATCGGAGCGAAAGGAAGCC	132
	Reverse: ACGTTGGTCACCGTTTTGTG	
*CYP76A2*	Forward: GCAGGTTTCACCGAGAGTGT	164
	Reverse: TGTTGCCTCTCCATCACACG	
*CYP77A3*	Forward: TTAGCAGTGCGGATTTGGCT	134
	Reverse: CGGACCGTAGAGTGAGGAGT	
**MYB transcription factor (4)**		
*RL1*	Forward: TTGAAAAGGCTCTGGCTGTGT	127
	Reverse: CTGATGTCTGCCACGAGGATT	
*RL6*	Forward: GCGTAACTGTGGCTCTACCT	102
	Reverse: GCTATGTTATGCCAGCGGTC	
*MYB114*	Forward: TTCGTAAGGGTGCATGGTGT	140
	Reverse: AAGCCACCTCAGTCTACAGC	
*MYB90*	Forward: AAAGGCACAAGCCTACCCTG	136
	Reverse: CTGGGGGCAGTGTCTTCATC	

## Data Availability

The datasets are publicly available at NCBI with Sequence Read Archive (SRA) accession: SRR16993328 to SRR16993332.

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
