# Peer review of "Integrated Metabolomic and Transcriptomic Analysis Reveals Differential Mechanism of Flavonoid Biosynthesis in Two Cultivars of Angelica sinensis"

_molecules, 2022, doi:10.3390/molecules27010306_

Round 1

Reviewer 1 Report

Dear authors,

Your manuscript is very rich in data and builds up its flow of argumentation well. Some things to consider when revising your manuscript are:

  1. There are quite a lot of English language flaws. Some of them were captured by me and are listed below, but this list is likely not exhaustive and help from a professional editor or native English speaker should be sought:

Keywords: "Metabolomic" (m instead of l), but note that rarely are adjectives given as keywords; you should pick nouns as keywords, for example "metabolomics" and "transcriptomics"

Throughout the manuscript: What do you mean by "violate" oils? Did you actually mean "volatile" oils? If so, please correct throughout the manuscript.

Page 1, line 40: roots WERE first recorded

Page 2, lines 55 and 79: no "of" between "among" and "the"

Page 2, line 68: 2.1. Difference in Flavonoid and Anthocyanin Contents... (there is no "s" at the end of those adjectival nouns)

Page 10, line 234: there should be no dash ("-") between f and l

Page 11, lines 291 and 308: cross out "into"

Page 11, lines 303 and 310: Absorbance readings were taken...

Page 11, line 315: "in" is missing before "a mixer mill"

Page 11, Line 324: "made up" is a better word choice than "were"

Page 14, line 391: should read "were greater than in M2"

2. Figure 1B: What is meant by "units" on the Y-axis?

3. There are so many acronyms used throughout the manuscript. I am well aware this is the norm in genomics/transcriptomics/metabolomics, but for some readers of Molecules this type of jargon may be less accessible. A centralized list of abbreviations would help keep track of them (including explanation of FDR, FC, etc.)

4. Standards that were used for some metabolite identifications are mentioned in section 4.3.4. Please detail what these standards were (source/manufacturer, purity, additional purification method, if performed) in a separate section called "Chemicals". Also include in this section the same details for all other reagents used in various experimental protocols: ethanol, methanol, NaNO2, AlCl3, Trizol, etc.  

Author Response

  1. There are quite a lot of English language flaws. Some of them were captured by me and are listed below, but this list is likely not exhaustive and help from a professional editor or native English speaker should be sought:

Keywords: "Metabolomic" (m instead of l), but note that rarely are adjectives given as keywords; you should pick nouns as keywords, for example "metabolomics" and "transcriptomics"

Thanks for your kind review, the keyword “Metabololic” has been revised to “Metabolomics”, and “Transcriptomic” has been revised to “Transcriptomics”. (Page 1, Line 32)

Throughout the manuscript: What do you mean by "violate" oils? Did you actually mean "volatile" oils? If so, please correct throughout the manuscript.

The words “violate” has been written by mistake, now it has been corrected to “volatile” throughout the manuscript. (Pages 1 and 2, Lines 13 and 46)

Page 1, line 40: roots WERE first recorded

According to your comments, the description “roots was first recoded” has been revised to “roots were first recorded”. (Page 1, Line 39)

Page 2, lines 55 and 79: no "of" between "among" and "the"

The word “of” has been deleted. (Page 2, Lines 54 and 79)

Page 2, line 68: 2.1. Difference in Flavonoid and Anthocyanin Contents... (there is no "s" at the end of those adjectival nouns)

The “s” at the end of “Flavonoids and Anthocyanins” has been deleted. (Page 2, Lines 67 and 68)

Page 10, line 234: there should be no dash ("-") between f and l

The dash "-" in the word “dihydroflavonol” has been deleted. (Page 10, Line 234)

Page 11, lines 291 and 308: cross out "into"

The word "into" has been deleted. (Page 11, Lines 303 and 314)

Page 11, lines 303 and 310: Absorbance readings were taken...

The description “Absorbance reader was taken...” has been revised to “Absorbance readings were taken...”. (Page 11, Lines 309, 316 and 317)

Page 11, line 315: "in" is missing before "a mixer mill"

The word “in” has been added in the sentence. (Page 11, Line 321)

Page 11, Line 324: "made up" is a better word choice than "were"

Thanks for your suggestion, the word “were” has been replaced by “made up”. (Page 12, Line 330)

Page 14, line 391: should read "were greater than in M2"

The description “is greater than M2” has been revised to “were greater than in M2”. (Page 14, Line 395)

  1. Figure 1B: What is meant by "units" on the Y-axis?

The units “μg” has been added on the Y-axis in the Figure 1B. (Page 2, Line 71)

  1. There are so many acronyms used throughout the manuscript. I am well aware this is the norm in genomics/transcriptomics/metabolomics, but for some readers of Molecules this type of jargon may be less accessible. A centralized list of abbreviations would help keep track of them (including explanation of FDR, FC, etc.)

According to your comments, Abbreviations have added in the text. (Pages 14 and 15, line 421)

Abbreviations

DAFs

differentially accumulated flavonoids

DEGs

differentially expressed genes

DR

down-regulated

FLNC

full-length non-chimeric

GO

Gene Ontology

KEGG

Kyoto Encyclopedia of Genes and Genomes

KOG

euKaryotic orthologous groups of proteins

M1

Mingui 1

M2

Mingui 2

NCBI

National Center for Biotechnology Information

OPLS-DA

orthogonal projection to latent structures-discriminant analysis

PCA

principal component analysis

REL

relative expression level

RPKM

Reads Per kb per Million

TF

transcription factor

UR

up-regulated

  1. Standards that were used for some metabolite identifications are mentioned in section 4.3.4. Please detail what these standards were (source/manufacturer, purity, additional purification method, if performed) in a separate section called "Chemicals". Also include 5in this section the same details for all other reagents used in various experimental protocols: ethanol, methanol, NaNO2, AlCl3, Trizol, etc.

According to your comments, the information of standards and normal reagents has been added: “4.2. Chemicals: Standards of metabolites used for UPLC analysis were purchased from BioBioPha, and Sigma-Aldrich, USA. All chemicals and reagents (e.g. AlCl3, catechin, ethanol, HCL, methanol, NaNO2, NaOH) were of analytical grade and purchased from Merck, Germany. Trizol reagent, RT Kit and SuperReal PreMix were purchased from Tiangen, China”. (Page 11, lines 295-300)

Reviewer 2 Report

The authors use a multi-omics approach to address mainly the flavonoids biosynthesis in two cultivars of Angelina sinensis. The experiments are correctly performed, and the data obtained from this study are a novel source of information on the metabolome and transcriptome of this species. Nevertheless, some modifications are needed and are listed below.

A critical point represents a weakness of the study: the samples analyzed are only leaves and petioles. It would have made more sense to include the stems since the authors say multiple times that the color of the stem is one of the main phenotypic differences between M1 and M2, and they also attribute some DAFs to this trait. However, despite this flaw, the data produced in this study are valuable for the scientific community.

Finally, it would be appropriate to perform English proofreading to fix some grammatical errors.

Line 13: I do not understand the meaning of “violate oils”. Do you mean volatile oils? Please fix this.

Line 16-17: The authors mention differences in yield and ferulic acid. It would be more precise to mention which cultivar has the higher yield and higher ferulic acid content. Just mentioning that there are differences sounds unclear, and it does not take more than a couple of words to be more precise. Please fix this.

Line 70: Please state which statistical test was used to detect the significant differences reported in figure 1.

Line 86: In table 1, the authors provide the log fold change but do not specify the set p-value and do not specify if the correction for multiple testing was part of the data analysis. This is crucial for the reliability of the reported results.

Line 118: The authors write that they had ~38 million reads on both cultivars after several filtering steps. What is missing is the information on the number of replicates. This is another crucial point on which depends the reliability of the reported results.

Line 212: The word “component” is too generalistic. Everything is a component. Please remove this sentence or rephrase it.

Line 292: Why did the authors use leaves and petioles and did not analyze stems instead? After all, stem color is one of the main phenotypic differences between M1 and M2.

Line 386: The number of replicates and the statistical test is appropriate, but they must be briefly mentioned also in the results section.

Author Response

Line 13: I do not understand the meaning of “violate oils”. Do you mean volatile oils? Please fix this.

The words “violate” has been written by mistake, now it has been corrected to “volatile” throughout the manuscript. (Pages 1 and 2, Lines 13 and 46)

Line 16-17: The authors mention differences in yield and ferulic acid. It would be more precise to mention which cultivar has the higher yield and higher ferulic acid content. Just mentioning that there are differences sounds unclear, and it does not take more than a couple of words to be more precise. Please fix this.

Thanks for your suggestion, the sentence has been revised to: “Although a higher root yield and ferulic acid content in M1 than M2 have been observed”. (Page 1, Line 16)

Line 70: Please state which statistical test was used to detect the significant differences reported in figure 1.

According to your comments, the method of statistical test has been added: “A t-test was performed for independent treatments.” (Page 2, Lines 72)

Line 86: In table 1, the authors provide the log fold change but do not specify the set p-value and do not specify if the correction for multiple testing was part of the data analysis. This is crucial for the reliability of the reported results.

According to your comments, the data analysis has been provided in the footnote in Table 1: “Note: The level of differential accumulation between M1 and M2 was determined with a criterion of variable importance in projection (VIP) ≥ 1 and t-test p ≤ 0.05.”. (Page 3, Lines 89-90)

Line 118: The authors write that they had ~38 million reads on both cultivars after several filtering steps. What is missing is the information on the number of replicates. This is another crucial point on which depends the reliability of the reported results.

According to your comments, the information on the number of replicates “(mean ± SD, n=3)” and the SD values have been added in the Table 1. (Page 3, Lines 86)

Line 212: The word “component” is too generalistic. Everything is a component. Please remove this sentence or rephrase it.

According to your comments, the sentence associated with “component” has been removed. (Page 9, Line 213)

Line 292: Why did the authors use leaves and petioles and did not analyze stems instead? After all, stem color is one of the main phenotypic differences between M1 and M2.

Indeed, you are providing a good question. In this study, the leaves and petioles were collected to perform the RNA sequencing and metabolites detection, there are three reasons: 1) almost of the metabolites are synthesized in the functional leaves, and transported through the petioles to the stem and others organs or tissues; 2) actually, the color of petioles is the same as the stem with purple; 3) it is difficult to extract the whole RNA samples due to abundant phenolic compounds, which induce the RNA degradation.

Line 386: The number of replicates and the statistical test is appropriate, but they must be briefly mentioned also in the results section.

Thanks for your suggestion, the number of replicates and the statistical test has been added accordingly in the Result section. (Pages 2, 3, 5 and 9, Lines 72, 86, 87, 123 and 205)